# Music Therapy as a Form of Nonpharmacologic Pain Modulation in Patients with Cancer: A Systematic Review of the Current Literature

**DOI:** 10.3390/cancers14184416

**Published:** 2022-09-11

**Authors:** Christopher Rennie, Dylan S. Irvine, Evan Huang, Jeffrey Huang

**Affiliations:** 1Department of Osteopathic Medicine, Nova Southeastern University Dr. Kiran C. Patel College of Osteopathic Medicine, Clearwater, FL 33759, USA; 2Department of Osteopathic Medicine, Nova Southeastern University Dr. Kiran C. Patel College of Osteopathic Medicine, Davie, FL 33314, USA; 3Carrollwood Day School, Tampa, FL 33613, USA; 4Department of Anesthesiology, Moffitt Cancer Center, Tampa, FL 33612, USA

**Keywords:** cancer, music therapy, pain modulation, pain management

## Abstract

**Simple Summary:**

Cancer is a condition that affects millions of people worldwide each year. Treatments include pharmacologic and surgical interventions that can pose great risks to the physical and mental health of patients. The objective of this systematic review is to consolidate the literature surrounding the use of music therapy as a low-risk and effective pain management adjunct to traditional cancer therapy. This analysis reveals that the use of music therapy thus far has provided a nearly unanimous positive effect on cancer patients, with the potential to provide both physical and psychosocial benefits. The apparent adverse effects appear to be negligible, and music therapy should be considered when creating a cancer care plan.

**Abstract:**

Aims and Objectives: To consolidate and summarize the current literature surrounding the use of music therapy as an effective noninvasive adjunct to conventional cancer therapy, especially as a low-risk alternative for pain management and anesthetic use in cancer patients. Background: Current studies have proposed that music therapy may be effective as a noninvasive adjunct to conventional cancer therapy in managing numerous outcomes in cancer patients. However, the findings of these investigations have not been consolidated and analyzed on a large scale. Therefore, focusing a systematic review on the effects of music therapy as an adjunct to conventional cancer therapy would give a better understanding of which intervention approaches are associated with better clinical outcomes for cancer patients. Design: A systematic review. Methods: A review of randomized controlled trials to evaluate the effectiveness of music therapy in physical, cognitive, and psychosocial outcomes for cancer patients alone or in conjunction with standard therapy was implemented. We conducted searches using the PubMed/MEDLINE, CINAHL, and Cochrane Library databases for all articles meeting the search criteria up until the time of article extraction in May, 2022. Only studies published in English were included. Two reviewers independently extracted data on participant and intervention characteristics. The main outcome variables included pain, anxiety, quality of life, mood, sleep disorders, fatigue, heart rate, blood pressure, respiratory rate, and oxygen saturation. Results: Of the 202 initially identified articles, 25 randomized controlled trials met the inclusion criteria for evaluation. Of the 25 studies, 23 (92.0%) reported statistically and clinically significant improvements across the outcome variables. Two of the studies (8.00%) found no significant positive effect from music therapy in any of the aforementioned outcomes variables. Conclusion: Music therapy, both as a standalone treatment and when used in conjunction with other pharmacologic and nonpharmacologic modalities, has a generally beneficial effect across several physiologic and psychosocial aspects of cancer.

## 1. Introduction

A new cancer diagnosis is a dreaded reality that many must face in their lifetime, with the prospect of chemotherapy, radiation, surgery, and even possible mortality in mind. In 2021, within the United States (U.S.) alone, an estimated 1.9 million people began to face this reality, and approximately 39.2% of men and women in the country will be diagnosed with cancer at some point in their lifetime [1]. While the general cancer 5-year survival rate has increased to 67.7%, the estimated number of cancer deaths in 2021 still reached slightly over 600,000, or roughly 1650 deaths per day [1,2].

While these metrics may be ominous, cancer mortality rates over the last 3 decades have dropped roughly 30% [1]. This is largely attributable to nationwide campaigns promoting smoking awareness and subsequent smoking cessation; however, expanded pre-screenings, genetic testing, and advanced medical technology have also played a role [1,3]. As a result, the rate of new cancer diagnoses has been increasing at a proportional rate. The American Cancer Society estimates the top five most common new cancer diagnoses in men to be prostate, lung, colorectal, bladder, and melanoma. These cancers make up nearly 60% of all new cancer diagnoses (roughly 600,000 cases) for men in the U.S. For women, the top 5 most common new cancer diagnoses include breast, lung, colorectal, uterine, and melanoma, also accounting for roughly 60% (550,000 cases) of new diagnoses [1].

The prevalence of each of these cancers varies widely, as both genetic predisposition and epigenetic factors influence rates. While familial history and genetic alterations are consistently implicated in cancer incidence, environmental risks and lifestyle behaviors such as poor diet, drinking alcohol, and obesity have become more prominent causative factors [4,5,6]. Other predisposing conditions can include viral infections such as Epstein–Barr Virus (EBV), ultraviolet (UV) radiation exposure, medications, occupational exposures, and chromosomal abnormalities [4,7,8]. These listed risks reflect only a small sample of the overarching complexity in cancer development, as each form of cancer is associated with both common and unique environmental and genetic considerations.

With both the increasing number of cancers and the evolution of cancer understanding, diagnostics have also progressed. Colonoscopies have become commonplace in American healthcare, aiding in the early detection and diagnosis of colorectal cancer [9,10]. Tumors of virtually any location can be noninvasively analyzed for malignancy with positron emission tomography (PET) scans, which show the rate of glucose uptake within a potentially cancerous neoplasm [11]. Some nations even offer this form of screening as a full-body scan for every individual over a certain age to obtain a baseline [12,13]. Varying methods of obtaining biopsies are now available as well, allowing for diagnosis of various growths in a relatively noninvasive manner [14].

Following a formal diagnosis, a range of treatment options are discussed. Often, some form of combination therapy is prescribed. The type, staging, and location of the cancer are all key components that play a role in the subsequent course of action with regard to the level of urgency and risk involved [15]. Some management modalities include surgical resection and excision, transplant surgery, radiation, chemotherapy, and immunotherapy [1,16].

Cancer patients can develop significant psychological distress, including anxiety, depression, post-traumatic stress symptoms, fear of cancer recurrence, pain, fatigue, and sleep disturbances [17]. This symptomology extends beyond the physical impact of a cancer diagnosis and must be addressed and treated accordingly [18]. The aforementioned methods are often the first thought of when a cancer diagnosis has been delivered; however, there is literature on further nontraditional methods such as music therapy (MT) for both the physical and mental rigors of such an illness.

MT can be utilized in numerous forms—a patient may listen to selected recordings or a live performance by a musical therapist. In addition to passive listening, patients may also actively participate within this setting and join in creating music. This form of outlet has been shown through numerous studies to aid in both the emotional and physical strains experienced by cancer patients [19,20,21,22]. MT has been used previously as an adjunctive therapy in the treatment diseases other than cancer [22,23]. These include, but are not limited to, the fields of psychiatry, general medicine, neurology, learning disabilities, and cardiovascular disease [22,23].

Studies have shown a significant reduction in pain, pharmacologic intervention, and emotional suffering as a result of MT [19,20,21,22,23]. The rationale stems from the ability to generate emotional and subsequent physical relaxation in response to the listening or creation of music, a concept that transcends medicine and is evident both currently and historically [23]. This form of intervention, whether a solitary treatment or as an adjunct, can present an option to reduce suffering in cancer patients while allowing for maintenance of baseline physiological and mental function [19,20,21,22,23].

The purpose of this review is to consolidate and summarize the current literature surrounding the use of music as an adjunct noninvasive cancer therapy. Our primary objective was to highlight MT as an effective noninvasive component to traditional combination treatment, especially as a low-risk alternative for pain management and anesthetic use in cancer patients [11,13].

## 2. Materials and Methods

For the purposes of this review, we sourced studies regarding MT as an adjunct treatment for cancer as it relates to both physical and emotional pain. This search was performed by two investigators in the following databases: PubMed/MEDLINE, CINAHL, and the Cochrane Library.

The search terms used in this literature review were as follows: “cancer” or “cancer patients” or “leukemia” or “neoplasm” or “tumor” AND “pain control” or “palliative care” AND “singing” or “choir” or “drumming” or “music” or “music therapy.”

The search sequence used in this review utilized the PICOS format of Patient, Interventions, Comparative Interventions, Outcomes, and Studies:P—Patients with cancerI—Received music therapy in addition to traditional cancer treatmentC—Received traditional cancer treatmentO—Pain, anxiety, quality of life, mood, sleep disorders, fatigue, heart rate (HR), blood pressure (BP), respiratory rate (RR), and oxygen saturation. S—Randomized controlled trials (RCTs)

This literature search was performed in May 2022. Only studies published or available in English were considered in this review. Inclusion criteria limited these articles to both observational studies and randomized controlled studies that focused on the use of MT for cancer pain management. Studies were not limited by a timeframe of publication, patient age, cancer type, or solitary MT. We included all studies where MT was utilized as a part of the cancer care plan, both as a standalone treatment and when used in conjunction with other noninvasive or invasive forms of therapy.

The results obtained from the original database search were subsequently screened manually by the two investigators for the aforementioned inclusion criteria. Those studies that met the criteria but lacked substantial pertinent data were also excluded. After initial exclusion by two independent investigators, full texts of the studies were read and evaluated. Pertinent data were extracted from the selected studies if they continued to meet inclusion criteria upon further review. Both of the authors assessed the methodological study quality independently, and a consensus was reached on the articles to be included in this analysis. All the reviews were performed blind, and any discrepancies on inclusion were later discussed and agreed upon. The Cochrane risk-of-bias tool for randomized trials was used to score all the articles prior to including them in this study. This decision was made because all the studies included in our systematic review were randomized controlled trials (RCTs). The authors determined that only studies with a perceived “low risk of bias” would be included in this study.

The articles screened were recorded in Table 1 with the following categories: study, first author, year of publication, country, type of study design, patients, MT type and treatment methods, clinical outcome evaluated, and main results. “Type of therapy” was designated as music and/or mixed, with music being solitary MT and mixed being MT in conjunction with other modes of treatment such as aroma therapy. “Treatment methods” expands on this section and demonstrates the way in which the MT was applied. This unbiased screening further limited the resultant publications, and only those that were deemed eligible by concurrent agreement were recorded in Table 1 and utilized in the subsequent analysis. The primary outcome variables used for analysis of MT were pain, anxiety, quality of life, mood, sleep disorders, fatigue, heart rate (HR), blood pressure (BP), respiratory rate (RR), and oxygen saturation.

## 3. Results

As seen in Figure 1, a total of 202 articles were originally identified in the primary search. After a review and exclusions were made, the results were limited to 25 publications eligible for the following qualitative analyses. Table 1 depicts the data derived from each qualifying paper, with information regarding authorship and publication, country of origin, clinical setting, patient data, type of therapy, treatment methods, study design, and main findings.

**Figure 1 cancers-14-04416-f001:**
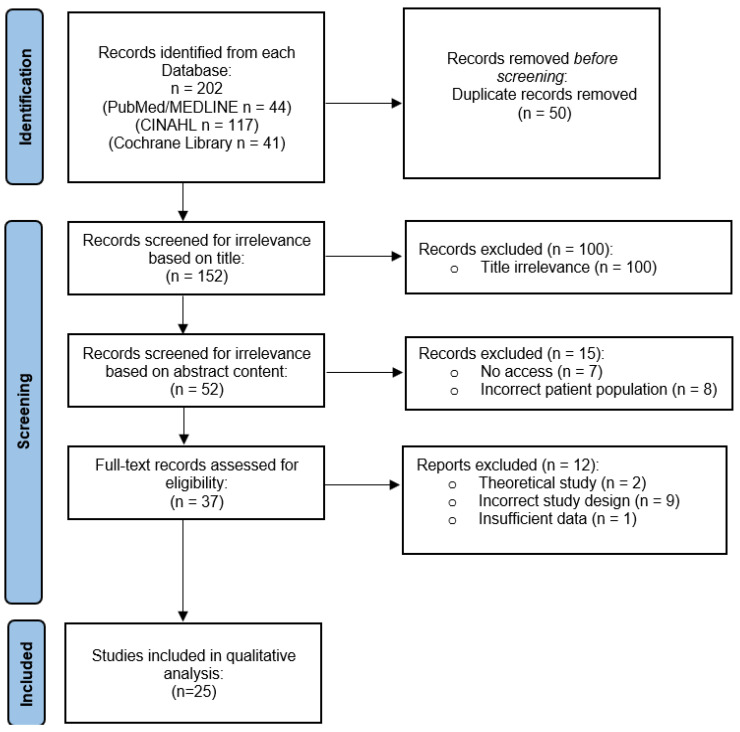
Prisma flow chart—identification of studies via databases.

**Table 1 cancers-14-04416-t001:** Summary of data extracted from relevant studies.

Study	Year	Country	Study Design	Patients	MT Type and Treatment Methods	Clinical Outcomes Evaluated	Main Results
Xiao et al. [24]	2018	China	RCT	N = 100 Tx. for breast CA, inpatient	MT and mixed therapy. Record 30 min prior to symptoms, 30 min after symptoms, 4 hours after symptoms	Stress and pain scores.	MT ↓ stress and pain in both tx. groups.
Liu et al. [25]	2019	China	RCT	N = 91 Tx. for osteo-sarcoma, in-patient	Mixed therapy. 30 min MT, followed by mindfulness-based stress reduction (MSBR), followed by 30 min of listening to any music	Pain, anxiety, and sleep dysfunction.	MT + MBSR ↓ pain, anxiety, and sleep disorders.
Nguyen et al. [26]	2010	Vietnam	RCT	N = 40 Tx. for leukemia, inpa-tient	MT. Received MT before, during, and after lumbar puncture (LP).	Anxiety, pain, RR, and HR.	MT ↓ anxiety and pain, RR, and HR.
Tang et al. [27]	2021	China	RCT	N = 100 Tx. for SCLC, inpatient	MT. 6 steps of MT before, during, and after chemotherapy.	Pain, anxiety, and sleep quality.	MT ↓ pain and anxiety, and ↑ sleep quality.
Reimnitz et al. [28]	2020	USA	RCT	N = 35 Tx. for blood and bone marrow CA, inpatient.	MT. Patient Preferred Live Music (PPLM).	Pain and fatigue.	PPLM MT ↓ pain and fa-tigue.
Warth et al. [29]	2016	Germany	RCT	N = 84 Tx. for un-known CA, inpatient.	MT. Live MT with pre and post therapy discussion, 30 min total.	Vascular sympathetic tone via stress and pain.	MT ↓ vascular sympathetic tone, stress, and pain
Tuinmann et al. [30]	2016	Germany	RCT	N = 66 Tx. for un-known CA, inpatient.	MT. Standard adjunct MT.	Pain, toxicity levels, and amount of antiemetics prescribed.	MT ↓ pain, tox-icities, and antiemetic use.
Kwekkeboom [31]	2007	USA	RCT	N = 60 Tx. for un-known CA, inpatient	MT. MT before and during procedure	Pain and anxiety scores.	MT showed no significant difference
Huang et al. [32]	2010	Taiwan	RCT	N = 126 Tx. for multiple CA types, inpatient	MT. Patients listened to a chosen recording out of four 60-80 bpm melodic tracks for 30 min.	Pain management compared to analgesic usage.	MT ↑ pain relief.
Shabanloei et al. [33]	2010	Iran	RCT	N = 100 Tx. for un-known CA, inpatient.	MT. MT during a bone marrow biop-sy/aspiration.	Pain and anxiety scores.	MT ↓ pain and anxiety.
Li et al. [34]	2011	China	RCT	N = 120 Tx. for breast CA, inpatient	MT. Patient preferred mu-sic via headphones 2x a day.	Short and long-term postoperative pain.	MT ↓ pain following mastec-tomy.
Deng et al. [35]	2021	China	RCT	N = 160 Tx. for breast CA, inpatient	MT and mixed therapy. MT +/- aroma therapy	Pain, anxiety scores, IL-6, and HMGB-1 levels.	MT +/– aroma therapy improved all out-comes.
Bieligmeyer et al. [36]	2019	Germany	RCT	N = 48 Tx. for multiple CA types, inpatient	MT. MT via vibroacoustic sound bed.	Emotional and physical well-being.	↑ Subjective emo-tional experiences and well-being.
Bradt et al. [37]	2015	USA	RCT	N = 31 Tx. for un-known CA, inpatient	MT and MM. 2 sessions of live music + 2 sessions of recorded music.	Pain and anxiety.	MT and MM ↓ pain and anxiety
Hsieh et al. [38]	2019	Taiwan	RCT	N = 60 Tx. for breast CA, at home	MT. Five 30 min sessions of HBMI for 24 weeks.	Mental fatigue, and pain intensity.	HBMI ↓ mental fatigue and pain intensity
Alam et al. [39]	2015	USA	RCT	N = 155 Tx. for skin CA, inpatient	MT. MT 4 days before and during surgery.	Intraoperative pain and anxiety.	MT showed no effect on intraoperative pain or anxiety
Hilliard [40]	2003	USA	RCT	N = 80 Tx. for multiple CA types, hospice	MT. Regular MT sessions.	Quality of life measures and lifespan.	Quality of life ↑. Lifespan had no change.
Bates et al. [41]	2015	USA	RCT	N = 108 Tx. for unknown CA, inpatient	MT. Two 30 min sessions 1 day prior and within 5 days of transplant.	Pain scores and amount of morphine required	MT ↓ pain and morphine use.
Wang et al. [42]	2015	China	RCT	N = 60 Tx. for lung CA, inpatient	MT. IV analgesia +/− MT pre- and post-op	VAS, SAS, BP, HR, analgesia frequency, and analgesic dosage	MT ↓ VAS, SAS, BP, HR, analgesia frequency, and dose
Burrai et al. [43]	2014	Italy	RCT	N = 52 Tx. for un-known CA, inpatient	MT. 30 minutes of live saxophone MT prior to treatment	Oxygen saturation and patient mood	MT ↑ oxygen saturation and mood
Clark et al. [44]	2006	USA	RCT	N = 63 Tx. for un-known CA, outpatient	MT. MT with preselected or self-selected music	Stress and anxiety	MT ↓ stress and anxiety
Walworth et al. [45]	2008	USA	RCT	N = 27 Tx. for brain CA, inpatient	MT. MT preoperatively and each day until discharge	Anxiety, relaxation, stress, and pre-procedure perception	MT ↑ quality of life measures. Length of stay was not affected
Bufalini [46]	2009	Italy	RCT	N = 39 Tx. for unknown CA, inpatient	MT. MT +/− conscious sedation	Conscious sedation, anxiety, and compliance	MT ↓ anxiety and ↑ compliance
Wint et al. [47]	2002	USA	RCT	N = 30 Tx. for unknown CA, inpatient	Mixed therapy. Patients undergoing LP were provided virtual reality (VR) glasses that incorporated visuals and music in a distraction therapy	Pain levels	VR glasses (with MT) ↓ pain
Ramirez et al. [48]	2018	Spain	RCT	N = 40 Tx. for un-known CA, inpatient	MT. One session of MT or company of the music therapist with no MT. EEG was performed for both groups	Emotional state, fatigue, anxiety, perceived ability to breathe, and relaxation	MT ↑ breathing ease and emotional state and ↓ fatigue and anxiety

Abbreviations: Blood Pressure BP, Cancer CA, Heart Rate HR, Home Based Music Intervention HBMI, Music Medicine MM, Music Therapy MT, Randomized Controlled Trial RCT, Respiratory Rate RR, Sedation Agitation Scale SAS, Treatment Tx, Visual Analog Scale VAS.

As part of the inclusion criteria, the 25 studies were all published or available in English and were performed in the following seven countries: USA (*n* = 9; 36.0%), China (*n* = 6; 24.0%), Germany (*n* = 3; 12.0%), Taiwan (*n* = 2; 8.00%), Italy (*n* = 2; 8.00%), Iran (*n* = 1; 4.00%), Spain (*n* = 1, 4.00%), and Vietnam (*n* = 1; 4.00%) [24,25,26,27,28,29,30,31,32,33,34,35,36,37,38,39,40,41,42,43,44,45,46,47,48]. The clinical setting for the administration of the MT was also recorded, with 22 (88.0%) studies occurring in inpatient hospitals [24,25,26,27,28,29,30,31,32,33,34,35,36,37,39,41,42,43,45,46,47,48], 1 (4.00%) in an outpatient facility [44], 1 (4.00%) in hospice care [40], and 1 (4.00%) as part of home healthcare [38].

In the 25 eligible records, 100% (*n* = 25) were randomized controlled trials (RCT), with a total of 1,875 cancer patients included. The number of participants included in each RCT ranged from 27 to 160, with the mean being 75.0 patients [24,25,26,27,28,29,30,31,32,33,34,35,36,37,38,39,40,41,42,43,44,45,46,47,48]. Demographic data such as patient age, sex, gender, and race were not reported in each eligible publication and thus were not included.

The criteria for this search allowed all types of cancer to be included. Eleven studies (44.0%) did not designate a specific form(s) of cancer being treated with MT and thus are listed as “Unknown” in Table 1 [26,29,30,31,33,37,39,41,43,44,46,47,48]. Three RCTs (12.0%) stated that MT was utilized for cancer patients, listing several types; these cases are represented in Table 1 with the “Multiple” designation [32,36,40]. The remaining studies each identified a single cancer type: breast (*n* = 4; 16.0%) [24,34,35,38], lung (*n* = 2; 8.00%) [27,42], osteosarcoma (*n* = 1; 4.00%) [25], leukemia (*n* = 1; 4.00%) [26], blood and marrow (*n* = 1; 4.00%) [28], skin (*n* = 1; 4.00%) [39], and brain (*n* = 1; 4.00%) [45].

Any form of music included within the care plan was considered for this literature search, regardless of whether it was a solitary treatment or part of a combination therapy. Of the 25 papers included in this qualitative analysis, 20 (80.0%) utilized MT alone as a single variable intervention [26,27,28,29,30,31,32,33,34,36,38,39,40,41,42,43,44,45,46,48]; two (8.00%) utilized mixed therapy [25,47], which in this case was a conjoined music-and-aroma treatment [24,35]; and two (8.00%) tested both music as a standalone and as part of an integrated therapy. The remaining study (*n* = 1; 4.00%) investigated the use of MT and music medicine each as single treatment components [37].

In the selected literature, the primary outcome variables for the MT consisted of improvements in pain, anxiety, quality of life, mood, sleep disorders, fatigue, heart rate, blood pressure, respiratory rate, and oxygen saturation [24,25,26,27,28,29,30,31,32,33,34,35,36,37,38,39,40,41,42,43,44,45,46,47,48]. Of the 25 studies, 22 (91.7%) reported statistically and clinically significant improvements across the parameters listed above, with the remaining 2 (8.00%) reporting no significant positive effect from the MT [24,25,26,27,28,29,30,32,33,34,35,36,37,38,40,41,42,43,44,45,46,47,48]. The 2 papers that studied solitary and combination MT found that while standalone MT provided significant improvements, mixed treatments provided even better results [24,35].

## 4. Discussion

Our review reveals a nearly unanimous positive benefit for MT across all physical, emotional, and mental parameters, including pain levels, anxiety, quality of life, mood and sleep disorders, fatigue, HR, BP, RR, and saturation [24,25,26,27,28,29,30,31,32,33,34,35,36,37,38,39,40,41,42,43,44,45,46,47,48]. The purpose of this review was to provide a systematic qualitative analysis of the current literature surrounding MT as a form of cancer treatment. MT was defined as the use of music or audio stimulation, regardless of form, administration, duration, or timing [49]. Studies including the use of MT alone or in conjunction with additional therapies were included as well. Records were not limited to forms of cancer or patient demographics such as age, race, sex, or gender. The results of this review are consistent with other studies [50,51,52].

92.0% of the selected articles found that MT, either alone or as part of a combination therapy, provided improvement or reduction in one or more of those categories. Interestingly, the two studies that had both an MT group and a combination therapy group found MT to be effective alone but more effective as part of a joint treatment, demonstrating a possible synergistic effect between these modalities [24,35]. Only one study investigated the use of MT with both its short- and long-term effects, demonstrating immediate rapid improvement in all forms of fatigue and progressively improving symptomology at 6, 12, and 24 weeks [38].

The anti-depressive and anti-anxiety effects of MT have shown to be useful for the mental rigors of illness, especially those associated with cancer [22]. Historically, MT has shown the capacity to elicit both positive and negative emotions via music association, ultimately leading to its perceived effects on fatigue, depression, anxiety, and pain [53]. The exact process by which this occurs is not fully understood, as music and its interpretation are highly personalized [53]. However, beyond the known psychosocial aspects that can be relieved with MT, there are a multitude of physical manifestations associated with cancer, many of which are affected not only by emotion, but also by the autonomic nervous system (ANS) [54]. Sympathetic nervous system (SNS) hyperactivity is highly implicated in the development, maintenance, and metastatic nature of neoplastic growth via the release of epinephrine, norepinephrine, and resultant glucocorticoid secretion [54]. Overstimulation of β-adrenergic receptors and glucocorticoid secretion through this mechanism creates a highly oncogenic environment, frequently increasing the incidence of tumor formation and spread [55,56]. Additionally, MT has become useful in the treatment of this pro-cancerous state via the induction of emotional response and SNS homeostasis regulation [55,56]. In this regard, MT can not only exert a beneficial effect emotionally, it can also positively impact the development, growth, and spread of cancer via music-mediated SNS downregulation [55,56].

As is often the case, there are inherent limitations in this analysis. The current literature available on the topic is limited, both in quantity and content. Of the studies included in this review, the average number of participants was 75.0, with a range of only 27 to 160 [24,25,26,27,28,29,30,31,32,33,34,35,36,37,38,39,40,41,42,43,44,45,46,47,48]. For all clinical trials, accurate calculation of a sample size is required for statistical and clinical significance, depending on a number of factors such as *p*-value, effect size, variance, and dropout rate [57]. With several studies holding such a small sample size, the ability to draw significance from the data is increasingly difficult. Additionally, there are limitations within the cancers represented in the included studies. Of the 25 studies, only 11 identified a single form of cancer, whereas the remaining 14 either treated patients with a list of multiple cancers or treated patients with cancer in general and did not make a designation of type. The most heavily represented cancers were breast and lung cancer, making up 16.0% and 8.00%, respectively. These cancers are both included in the top five most common cancers in the U.S. By contrast, the remaining forms presented in these studies can be categorized as rarer; these include brain cancer, which makes up less than 3% of cancer deaths in American men and women [1,58]. Moreover, while the study includes all MT aids in an attempt to create a broad recommendation for the effect of music in oncologic medicine, there are limitations in the generalizability of the data due to the wide variability in administration. This represents another point for improvement in the field regarding the most optimal techniques of MT and its applications. Patients may benefit from personalized music selections as opposed to preselected music, live music as opposed to recorded music, being an active participant in creating music, or from different forms of music such as instrumentals, vocals, and so on [59,60]. The possibilities of MT and music medicine are vast, not only in their array of uses but also in the patients themselves, who may have differing connections and experiences with music [14].

The literature search was limited to PubMed/MEDLINE, CINAHL, and the Cochrane Library, and studies had to have been published in English. It is highly likely that other otherwise suitable studies were not included.

Ultimately, these points serve to emphasize both the general recommendation of MT for cancer patients and the necessity for more research on the topic. There is sufficient, albeit limited, data available which we have shown to corroborate the validity of MT in treating the physical, emotional, and mental suffering associated with cancer. While this review helps add to the current literature on the topic, there remains a relative disparity in this data as compared to data on other forms of treatment. As seen in the initial query, it is evident that the literature on this topic is limited, and even more so when analyzing specific parameters. Knowledge on this topic can be further improved with in-depth analyses on music therapy across each specific form of cancer, different forms of music therapy (i.e., live, recorded, vocal, instrumental, combination), self-preferred versus preselected music, receptive versus active, and so on [14,59,60]. This review both reaffirms music therapy to be effective via a variety of measures, and emphasizes the potential for continuing studies to help us better understand the use of music therapy and help tailor treatments to the needs of cancer patients.

Although there are limitations to this review as acknowledged above, there are numerous points of strength as well. This review provides a comprehensive and robust evaluation of the current literature surrounding this topic. The literature search featured multiple databases, and the search sequence was intentionally broad so as to allow for a larger range of articles to be processed. The inclusion criteria allowed this review to analyze data with no constraint on age, geographic location, time of study, number of participants, type of cancer, stage of cancer, or type of music therapy performed. The exclusion methodology maintained the integrity of the analysis via a blind independent review and an article quality assessment. The data presented here can be reported with confidence and practical applicability, allowing us to emphasize a substantial recommendation for MT as a form of pain modulation in cancer patients.

## 5. Conclusions

Cancer is an illness that is physically and mentally taxing, both inherently and due to the treatments necessary for survival. The currently available therapies have advanced significantly in modern medicine; however, a majority of them pose substantial and oftentimes lethal risks. Here we have successfully synthesized the available data pertaining to the implementation of MT—a nonpharmacologic modality which we have shown to possess substantial psychosocial and physiologic benefits with minimal to no patient harm.

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
