# Peer review of "Music Therapy as a Form of Nonpharmacologic Pain Modulation in Patients with Cancer: A Systematic Review of the Current Literature"

_cancers, 2022, doi:10.3390/cancers14184416_

Round 1

Reviewer 1 Report

The purpose of this review was to provide a systematic qualitative analysis of the current literature on Music Therapy (MT) as a form of cancer treatment. Based on the extensive analysis of the literature, the Authors provided statistically significant data confirming that Music Therapy can improve both, physiologic and psychosocial aspects of cancer. 

This is a very interesting study confirming that MT may well complement adjuvant therapy. The Article can be published in Cancer.   There is only one small recommendation: 1. Please, provide a small chapter explaining the Nature of the positive effects of Music on recovery from cancer.   For this propose, it would be good to analyse literature concerning relationships between Music and autonomous nervous systems, particularly Sympathetic nervous system that promotes metastasis at hyperactivated state. (Music can potentially soothe the Sympathetic NS)   It also would be good to include the evidence about the anti-inflammatory and antidepressive  effects of music.  

Reviewer 2 Report

The manuscript was prepared very well. The introduction section justifies the purpose of the study. I congratulate the authors for the preparation of the manuscript

However, I have the following comments:

Introduction

·       Indicate whether music therapy (MT) has been used previously in other diseases, and for what indications.

·        Include what the TM consists of, and the existing application history. Are there several methodologies? Clarify this question

Methods

·       The search in a database is a bit limited, could you do it in at least 2 more? PubMed is a search engine, the base is Medline.

·       Please insert in Annexes the search sequence used with the key words you used.

·       I recommend that you register the review in PROSPERO.

·       I recommend using an evaluation of the methodological quality of the articles chosen: McMaster, PEDRo...

Results

·       In the tables of results regroup some columns and also reorder. add a more complete description of the patients and ellipsis of the results letter (use symbols)

·       Study, years and country; type of study, patients, MT and Treatment Methods, what is evaluated and results.

Discussion

·       Include a section on strengths.

·       What does this article contribute to, the authors should make their own assessment and include their own discussion of the results shown in the manuscript?

·       In the Conclusion section, state the most important outcome of your work. Do not simply summarize the points already made in the body — instead, interpret your findings at a higher level of abstraction. Show whether, or to what extent, you have succeeded in addressing the need stated in the Introduction (or objectives).

Round 2

Reviewer 2 Report

The problem with the manuscript is the tables, which should be improved with the incorporation of signs or symbols. They resemble the abstracts of the studies themselves.

The selected articles (n=25) should undergo a methodological quality assessment (McMaster or PEDRO).

Does not include the question PICOS

The review has not been registered in PROSPERO.

A search sequence with the words used for it is not included.

In general, some review comments have been omitted.
